# Operator entanglement in $SU(2)$-symmetric dissipative quantum many-body dynamics

**Lin Zhang**[1*]

1 ICFO-Institut de Ciencies Fotoniques, The Barcelona Institute of Science and Technology, Av. Carl Friedrich Gauss 3, 08860 Castelldefels (Barcelona), Spain

⋆ lin.zhang@icfo.eu

## Abstract

The presence of symmetries can lead to nontrivial dynamics of operator entanglement in open quantum many-body systems, which characterizes the cost of an matrix product operator (MPO) representation of the density matrix and provides a measure for the classical simulability. One example is the U(1)-symmetric open quantum systems with dephasing, in which the operator entanglement increases logarithmically at late times instead of being suppressed by the dephasing. Here we numerically study the far-from-equilibrium dynamics of operator entanglement in a dissipative quantum many-body system with the more complicated SU(2) symmetry and dissipations beyond dephasing. We show that after the initial rise and fall, the operator entanglement also increases again in a logarithmic manner at late times in the SU(2)-symmetric case. We find that this behavior can be fully understood from the corresponding U(1) subsymmetry by considering the symmetry-resolved operator entanglement. But unlike the U(1)-symmetric case with dephasing, both the classical Shannon entropy associated with the probabilities for the half system being in different symmetry sectors and the corresponding symmetry-resolved operator entanglement have nontrivial contributions to the late time logarithmic growth of operator entanglement. Our results show evidence that the logarithmic growth of operator entanglement at long times is a generic behavior of dissipative quantum many-body dynamics with U(1) as the symmetry or subsymmetry and for more broad dissipations beyond dephasing. We show that the latter is valid even for open quantum systems with only U(1) symmetry by breaking the SU(2) symmetry of our quantum dynamics to U(1).

## 1   Introduction

Recent years have witnessed remarkable experimental advances in atomic, molecular, and optical physics, which have enabled us to engineer quantum many-body systems in controllable and clean environments at the level of individual atoms, molecules, and ions [1–5]. These achievements provide us fantastic insights into the studies of strongly correlated quantum systems [6–9]. Among them, extensive attention has been focused on understanding quantum many-body dynamics far away from equilibrium [10, 11], an outstanding challenge in the modern physical sciences, and many new states of matter have been uncovered, such as the many-body localization [12,13], time crystals [14,15], and quantum many-body scars [16,17]. However, it is usually hard to characterize these nonequilibrium systems.

One important tool to describe the quantum many-body dynamics is the so-called entanglement entropy. On the one hand, its growth as a function of time can show fundamentally distinct behavior in different nonequilibrium systems. For instance, the growth of entanglement entropy is logarithmic in the many-body localized models [18], while it increases linearly for a generic chaotic system. This provides a defining feature for many nonequilibrium states of matter. On the other hand, the growth of entanglement entropy also provides insights into the classical simulations of quantum many-body dynamics using tensor networks [19–21]. In one dimension (1D), the quantum states can be faithfully represented by matrix-product states (MPSs) in terms of local three-rank tensors with bond dimension $\chi$ [22,23], for which the entanglement entropy is bounded by $\log_2 \chi$. Thus the linear increasement of entanglement entropy will lead to an exponential growth of bond dimension, which is considered to be extremely hard to simulate [24]. Only the quantum dynamics within a low-entanglement manifold of the Hilbert space can be efficiently simulated on classical computers. Thanks to the above reasons, studying the growth of entanglement entropy has become one of the central tasks in investigating quantum many-body dynamics and has attracted broad interest [25–28].

Not only being useful in studying the closed systems, the growth of entanglement entropy also helps the understanding of open quantum many-body systems, which are ubiquitous in practical experiments due to the inevitable couplings to environments. Like the MPS representation for pure states, the density matrix of 1D open quantum many-body systems can be described by the matrix product operators (MPOs) [29–31]. The bipartition of MPO through Schmidt decomposition further defines the entanglement entropy in operator space, which is dubbed as the operator entanglement and characterizes the cost of an MPO representation, i.e., how many Schmidt values are needed at least for faithfully representing an operator [32–37], hence providing a measure for the classical simulability of open quantum many-body systems.

In open quantum systems, after the initial linear growth reminiscent of unitary quantum dynamics, the operator entanglement is expected to be suppressed by the dissipations. However, it was recently studied that the operator entanglement in U(1)-symmetric open quantum many-body systems with dephasing increases logarithmically at long times [38], which is attributed to the growth of classical Shannon entropy associated with the probabilities for the

half system being in different U(1) sectors. This highlights the role of symmetries on the operator entanglement dynamics in open quantum many-body systems, which, however, still remains largely unexplored. One important symmetry is the non-Abelian SU(2) symmetry, which is believed to be crucial in many nonequilibrium phenomena like quantum many-body scars [39] and anomalous finite-temperature transport [40]. As the non-Abelian symmetry in general increases the entanglement entropy [41–43], it would be interesting to study how the long-time behavior of operator entanglement is impacted in the presence of SU(2) symmetry. Particularly, as the SU(2)-symmetric open quantum many-body system contains U(1) as the subsymmetry and dissipations beyond dephasing, it would be interesting to know whether the logarithmic growth of operator entanglement can go beyond open quantum systems with U(1) symmetry and dephasing.

In this work, we numerically study the far-from-equilibrium dynamics of operator entanglement in a dissipative quantum many-body system with SU(2) symmetry. We show that after the initial rise and fall, the operator entanglement also increases again in a logarithmic manner at long times in the SU(2)-symmetric case. We find that this behavior can be fully understood from the corresponding U(1) subsymmetry by considering the symmetry-resolved operator entanglement [44–46]. But unlike the U(1)-symmetric case, both the classical Shannon entropy and the symmetry-resolved operator entanglement now have nontrivial contributions to the late time logarithmic growth of operator entanglement. Our results show evidence that the logarithmic growth of operator entanglement at long times is a generic behavior of dissipative quantum many-body dynamics with U(1) as the symmetry or subsymmetry and for more broad dissipations beyond dephasing. We show the latter is even valid for open quantum systems with only U(1) symmetry by breaking the SU(2) symmetry of our quantum dynamics to U(1).

## 2 Model

We consider the open quantum many-body dynamics on an infinite spin-1/2 chain governed by the Lindblad master equation ($\hbar \equiv 1$)

$$\frac{\mathrm{d}}{\mathrm{d}t}\rho = -\mathrm{i}[H,\rho] + \gamma \sum_i \left( L_i \rho L_i^\dagger - \{L_i^\dagger L_i, \rho\}/2 \right) \equiv \mathcal{L}[\rho], \qquad (1)$$

where the Hamiltonian $H = J \sum_i P_{i,i+1}$ with the exchange operator $P_{i,i+1} = 2\mathbf{S}_i \cdot \mathbf{S}_{i+1} + 1/2$ is the 1D Heisenberg model with SU(2) symmetry. Here $\mathbf{S}_i$ denotes the spin-1/2 operator on site $i$, and $J$ is the nearest-neighbor spin coupling strength. We couple the system to an environment through the Lindblad operator $L_i = P_{i,i+1}$ with strength $\gamma$, which describes the dissipation proportional to the dipole interaction between neighbor sites; see Fig. 1(a) for the sketch. Similar model has been considered in Refs. [47,48] to study the stability or absence of superdiffusion in a spin chain with fluctuating exchange couplings that break the integrability. Since $L_i$ commutes with the total spin operator, the above Lindblad master equation preserves the SU(2) symmetry. To study the quantum dynamics with SU(2) symmetry, we consider the initial state $\rho_0 = |\psi_0\rangle\langle\psi_0|$ with $|\psi_0\rangle = \bigotimes_i (|\uparrow\rangle_{2i-1} |\downarrow\rangle_{2i} - |\downarrow\rangle_{2i-1} |\uparrow\rangle_{2i})/\sqrt{2}$, which is the product state of singlet pairs. This state has total spin 0 and is SU(2)-symmetric.

## 3 MPO decomposition of density matrix

To solve the Lindblad master equation, we use MPO to represent the density matrix. The matrix product decomposition of density matrix is a mixed-state version of the MPS, where the density matrix $\rho$ of a spin-1/2 chain with local Hilbert space dimension $d = 2$ is treated as

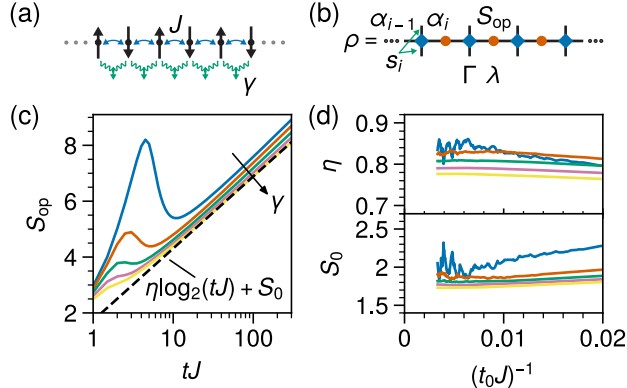

Figure 1: Operator entanglement dynamics in the SU(2)-symmetric dissipative quantum many-body system. (a) Sketch of the model. We consider a quantum spin chain with coherent nearest-neighbor coupling $J$ (blue arrows) and local dissipation proportional to the dipole interaction between neighbor sites with strength $\gamma$ (green arrows). (b) MPO decomposition of the density matrix $\rho$ in terms of local tensors $\Gamma$ (blue squares) and $\lambda$ (orange circles). Here $\alpha_{i-1}$ and $\alpha_i$ are the bond indices, while $s_i$ denotes the combined physical index of the bra and ket legs at site $i$. The operator entanglement $S_{\mathrm{op}}$ at certain bond can be calculated from the Schmidt vectors $\lambda$. (c) Time evolution of $S_{\mathrm{op}}$ for the product initial state of singlet pairs with different dissipation strength $\gamma = 0.05J$, $0.10J$, $0.15J$, $0.20J$, and $0.25J$. The black dashed line indicates the logarithmic growth of operator entanglement at long times (log-scale time axis), i.e., $S_{\mathrm{op}}(t \to \infty) = \eta \log_2(tJ) + S_0$. (d) Numerical prefactor $\eta$ and offset $S_0$ obtained from the local tangent of operator entanglement at time $t_0 J$. The results are converged for time step $\delta t J = 0.5$ and maximum bond dimension $\chi = 50000$.

a vector $|\rho\rangle_\sharp$ in the tensor product space of the $d \times d$ complex matrices [29]. Here we use the subscript $\sharp$ (sharp) to denote operators (superoperators) when represented as the superkets $|\cdot\rangle_\sharp$ (mappings between superkets), with $\langle s|s'\rangle_\sharp \equiv \mathrm{Tr}(s^\dagger s')/d$. Given a set of orthonormal basis $\{|s_i\rangle_\sharp\}$ for each site $i$, the density matrix $|\rho\rangle_\sharp$ can be decomposed as

$$|\rho\rangle_\sharp = \sum_{\{s_i\}} \sum_{\{\alpha_i\}} \prod_i \Gamma^{[i]s_i}_{\alpha_{i-1}\alpha_i} \lambda^{[i]}_{\alpha_i} \bigotimes_i |s_i\rangle_\sharp \tag{2}$$

through a succession of Schmidt decompositions [29], where $\Gamma^{[i]}$ is a three-rank tensor on site $i$ with one combined physical index $s_i$ and two bond indices $\alpha_{i-1}$, $\alpha_i$, while $\lambda^{[i]}$ is the Schmidt vector on bond $i$; see Fig. 1(b). Since both the initial state and Liouvillian operator are invariant under the translation by two sites, we have $\Gamma^{[i+2]} = \Gamma^{[i]}$ and $\lambda^{[i+2]} = \lambda^{[i]}$. Therefore, the numerical simulation can be performed within a unit cell of two sites for the infinite lattice, for which only two $\Gamma$ and $\lambda$ tensors are needed to capture the density matrix.

For the Liouvillian superoperator (1) that can be decomposed into terms involving at most two contiguous sites, the time evolution can be simulated using the infinite time-evolving block decimation (iTEBD) algorithm with a fourth-order Trotter decomposition of the matrix exponential of the Liouvillian $\exp(\mathcal{L}_\sharp \delta t)$ for a time step $\delta t$ [21]. We note that since the open quantum many-body dynamics governed by the Lindblad master equation is nonunitary, the canonical form of the infinite MPO is in general not preserved in the original iTEBD algorithm [49]. As a consequence, the truncation errors accumulate fast and ruin the simulation at long times. To overcome this issue, we employ the improved algorithm for nonunitary evolutions proposed in Ref. [50], in which the tensors updated in each time step are reorthonormalized into the canonical form. To avoid the exponential growth of bond dimension, we also

truncate the tensors $\{\Gamma^{[i]}\}$ and $\{\lambda^{[i]}\}$ at a maximum bond dimension $\chi$. The results shown in this work are all numerically converged.

## 4  Operator entanglement

We focus on the time evolution of operator entanglement in our SU(2)-symmetric dissipative quantum many-body dynamics. The operator entanglement is a basis-independent measure for quantum operators. Given the MPO decomposition (2), the operator entanglement at certain bond is defined as [32–37]

$$S_{\text{op}} \equiv -\sum_\alpha \lambda_\alpha^2 \log_2 \lambda_\alpha^2, \tag{3}$$

where the Schmidt values are assumed to be normalized, i.e., $\sum_\alpha \lambda_\alpha^2 = 1$. We note that the operator entanglement does not characterize the quantum entanglement between different parts of the system, which is instead captured by the entanglement measures like entanglement negativities when $\rho$ is a mixed state [51,52]. The operator entanglement mainly reflects the cost of encoding an operator in the MPO representation and also provides insights into nonequilibrium quantum many-body physics like the quantum chaos and information scrambling [53–55].

For our SU(2)-symmetric quantum dynamics, the total spin $\mathbf{S}_{\text{tot}} = \sum_i \mathbf{S}_i$ is conserved. Therefore, we have $\mathbf{S}_{\text{tot}}^2 \rho = S_{\text{tot}}(S_{\text{tot}} + 1)\rho$ during the time evolution, where $S_{\text{tot}}$ is zero for our initial state $\rho_0$ as a product state of singlet pairs. Consider a bipartitioning of the system, $\rho = \sum_\alpha \lambda_\alpha \varrho_\alpha^{[\text{L}]} \otimes \varrho_\alpha^{[\text{R}]}$, with $\langle \varrho_\alpha^{[\text{L/R}]} | \varrho_{\alpha'}^{[\text{L/R}]} \rangle_\sharp = \delta_{\alpha\alpha'}$ for the left (L) and right (R) part. We also have $\mathbf{S}_{\text{L/R}}^2 \varrho_\alpha^{[\text{L/R}]} = S_{\text{L/R}}(S_{\text{L/R}} + 1)\varrho_\alpha^{[\text{L/R}]}$ with $S_{\text{L}} = S_{\text{R}} \equiv S$, where $\mathbf{S}_{\text{L/R}}$ is the total spin operator in the left/right part of the system. For this, we can relabel the bond index $\alpha$ as $\alpha \to (S, i_S)$, where $i_S$ distinguishes the Schmidt values (i.e., different half-system density matrices) corresponding to the same half-system total spin $S$. We note that in our notation the degenerate degrees of freedom of each spin sector $S$ is not counted into $i_S$. For each bond label $(S, i_S)$, there are actually $(2S + 1)$ Schmidt coefficients with the same value $\lambda_{(S,i_S)}$ in our numerical simulation with SU(2) symmetry. Hence the operator entanglement is given by $S_{\text{op}} = -\sum_S \sum_{i_S}(2S + 1)\lambda_{(S,i_S)}^2 \log_2 \lambda_{(S,i_S)}^2$ in this notation with the normalization condition $\sum_S \sum_{i_S}(2S + 1)\lambda_{(S,i_S)}^2 = 1$. We also remark that the quantum number $S$ can also be defined by multiplying the symmetry operator from the right of density matrix, which does not change the results.

It is useful to also introduce the symmetry-resolved operator entanglement [44–46], which has attracted extensive attention recently [56–61]. To this end, we define the probability of having total spin $S$ in the half system as $p_S = (2S + 1)\sum_{i_S} \lambda_{(S,i_S)}^2$. Then the symmetry-resolved operator entanglement in the spin sector $S$ is given by $S_{\text{op},S} = -(2S + 1)\sum_{i_S} \hat{\lambda}_{(S,i_S)}^2 \log_2 \hat{\lambda}_{(S,i_S)}^2$ with $\hat{\lambda}_{(S,i_S)} \equiv \lambda_{(S,i_S)}/\sqrt{p_S}$. With this, the full operator entanglement can be recast as

$$S_{\text{op}} = \sum_S p_S S_{\text{op},S} - \sum_S p_S \log_2 p_S, \tag{4}$$

where the first term describes the averaged contribution from the symmetry-resolved operator entanglement in different spin sectors, while the second term is the classical Shannon entropy of the probability distribution.

Since our model has the U(1) subsymmetry, e.g., the conservation of total magnetization along the $z$ direction, we can also label the Schmidt coefficients via the half-system magnetization and define the corresponding symmetry-resolved operator entanglement; see Refs. [38,

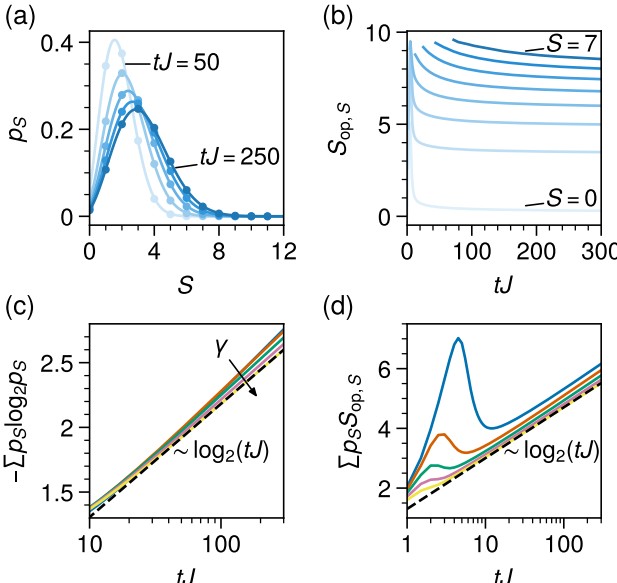

Figure 2: Symmetry-resolved operator entanglement in spin sectors. (a) Probabilities $p_S$ of the infinite chain with total spin $S$ in the half system at increasingly late times ($50 \leq tJ \leq 250$ from light to dark) for $\gamma = 0.05J$. The lines are fits with the trial function shown in the main text. (b) The corresponding symmetry-resolved operator entanglement $S_{\text{op},S}$ as a function of time. Only the data with probabilities $p_S > 10^{-4}$ are presented. (c) (d) The classical Shannon entropy $-\sum p_S \log_2 p_S$ and the averaged symmetry-resolved operator entanglement $\sum p_S S_{\text{op},S}$ as a function of time for $\gamma/J = 0.05, 0.10, 0.15, 0.20,$ and $0.25$. The black dashed lines indicate the logarithmic growth at late times (log-scale time axis). The results are converged for the time step $\delta tJ = 0.5$ and maximum bond dimension $\chi = 50000$.

59, 61]. For our SU(2)-symmetric case, since each spin sector with $S \geq |S_z|$ contributes one state to the magnetization sector $S_z$, we can express the probability of having magnetization $S_z$ in the half system as $p_{S_z} = \sum_{S \geq |S_z|} \sum_{i_S} \lambda^2_{(S,i_S)}$, and the symmetry-resolved operator entanglement in this magnetization sector is given by $S_{\text{op},S_z} = -\sum_{S \geq |S_z|} \sum_{i_S} \tilde{\lambda}^2_{(S,i_S)} \log_2 \tilde{\lambda}^2_{(S,i_S)}$ with $\tilde{\lambda}_{(S,i_S)} \equiv \lambda_{(S,i_S)} / \sqrt{p_{S_z}}$. The relation to the full operator entanglement $S_{\text{op}}$ is similar to Eq. (4).

## 5 Results

### 5.1 Logarithmic growth of operator entanglement

We now present the numerical results for our open quantum many-body system (1). In Fig. 1(c), we show the time evolution of operator entanglement $S_{\text{op}}$ for the product initial state of singlet pairs. Initially, the operator entanglement grows linearly in time, as the quantum dynamics at such short times is dominated by the Hamiltonian part of Eq. (1) and can be approximated to be unitary. However, when $t \gtrsim \gamma^{-1}/4$ in our numerical simulation, the coupling to the environment becomes relevant, and the operator entanglement starts to decrease.

Typically, if there is no conservation law in the dissipative quantum many-body dynamics, the density matrix is expected to relax to the infinite-temperature state (i.e., the identity matrix), and the operator entanglement converges toward the corresponding value of that stationary state at late times. However, the introduction of symmetries can enrich the behavior of op-

erator entanglement. Particularly, it was recently reported that after the initial rise and fall, the operator entanglement in U(1)-symmetric open quantum many-body systems with dephasing increases again in a logarithmic manner at late times, i.e., $S_{op}(t \to \infty) = \eta \log_2(tJ) + S_0$ [38].

Here we also observe the same behavior of operator entanglement in our dissipative quantum many-body dynamics with SU(2) symmetry; see Fig. 1(c). Especially, we show the prefactor $\eta$ and offset $S_0$ as a function of time $t_0$ in Fig. 1(d), which is obtained as the local tangent of operator entanglement. Due to the limited maximum bond dimension, the data for small $\gamma$ is not good enough as those for the strongly dissipative cases. However, all of them show the tendency to converge to a finite value as $t_0 \to \infty$, thus identifying the logarithmic growth behavior of operator entanglement at late times. We note that unlike the U(1)-symmetric case with dephasing [38], here both the prefactor $\eta$ and offset $S_0$ are nonuniversal and depends on the value of the dissipation strength.

## 5.2 Symmetry-resolved operator entanglement in spin sectors

To understand the logarithmic growth behavior of operator entanglement in the quantum dynamics with SU(2) symmetry, it is useful to consider the symmetry-resolved operator entanglement. We first consider the spin sectors. We show the probabilities $p_S$ of having total spin $S$ in the half system at late times in Fig. 2(a). The half-system total spin is mainly distributed in the low-spin sectors, but as time increases the system has more probabilities in the high-spin sectors. We fit the probability distribution with the trial function

$$p_S = \frac{2S+1}{\sqrt{2\pi\delta^2}} \left[ e^{-S^2/2\delta^2} - e^{-(S+1)^2/2\delta^2} \right] \tag{5}$$

with parameter $\delta$ and find a good match [see lines in Fig. 2(a)]. Considering the relation $p_S = (2S+1)(p_{S_z=S} - p_{S_z=S+1})$, this suggests that the probability of having magnetization $S_z$ in the half system follows the Gaussian distribution with variance $\delta$ in our dissipative quantum many-body dynamics with SU(2) symmetry, like the one observed in the U(1)-symmetric case with dephasing [38].

We also present the symmetry-resolved operator entanglement in different spin sectors in Fig. 2(b). Unlike Ref. [38], where the symmetry-resolved operator entanglement drops to very small values quickly, here the operator entanglement $S_{op,S}$ remains finite and large at late times, indicating that in addition to the classical Shannon entropy of probability distributions, the symmetry-resolved operator entanglement also have nontrivial contributions to the logarithmic growth of the total operator entanglement in our SU(2)-symmetric case, as we show in Figs. 2(c) and 2(d).

## 5.3 Symmetry-resolved operator entanglement in magnetization sectors

Despite the different behavior of symmetry-resolved operator entanglement at late times, the analysis of probabilities $p_S$ suggests that the logarithmic growth behavior of operator entanglement in our dissipative quantum many-body dynamics with SU(2) symmetry can also be understood from the corresponding U(1) subsymmetry by considering the symmetry-resolved operator entanglement in magnetization sectors.

As we expected, the probabilities $p_{S_z}$ indeed are approximately Gaussian at late times, i.e., $p_{S_z} \simeq e^{-S_z^2/2\delta^2}/\sqrt{2\pi\delta^2}$; see Fig. 3(a). In Fig. 3(b), we also present the time dependence of the variance $\delta$, which follows the power law $\sim (tJ)^\alpha$ at late times and explains the logarithmic growth behavior of $-\sum_S p_S \log_2 p_S$ shown in Fig. 2(c). The exponent $\alpha$ in general depends on the dissipation strength and for large $\gamma$ is close to the value 0.25 predicted for the U(1)-symmetric dissipative quantum dynamics with dephasing [38]; see the insert in

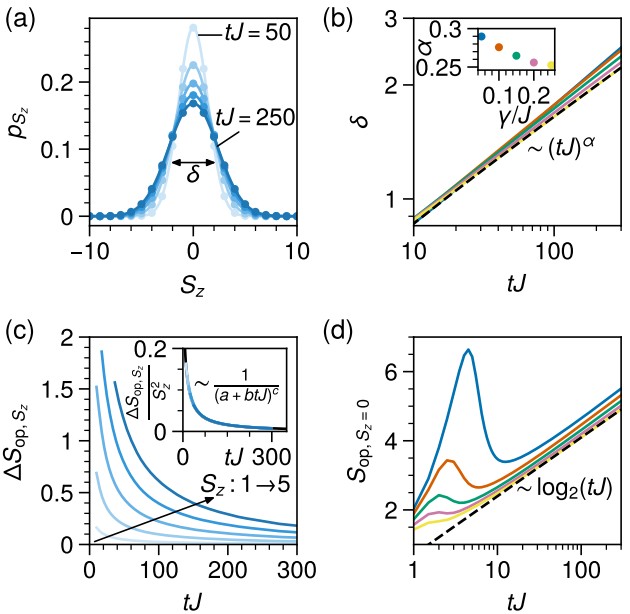

Figure 3: Symmetry-resolved operator entanglement in magnetization sectors. (a) Probabilities $p_{S_z}$ of the infinite chain with magnetization $S_z$ in the half system at increasingly late times ($50 \leq tJ \leq 250$ from light to dark) for $\gamma = 0.05J$. Lines are the Gaussian fits. (b) Variance $\delta$ of the Gaussian fits as a function of time for $\gamma/J = 0.05, 0.10, 0.15, 0.20$, and $0.25$. The black dashed line indicates $\delta \sim (tJ)^\alpha$ at late times (double-log scale). The corresponding exponent $\alpha$ for each $\gamma$ is shown in the insert. (c) Symmetry-resolved operator entanglement difference $\Delta S_{\text{op},S_z}$ as a function of time for $\gamma = 0.05J$ and $S_z = 1, 2, 3, 4, 5$ (light to dark). The insert shows the data scaled by $1/S_z^2$, and the black line is the fit $(a + btJ)^{-c}$ with $a = 2.4964$, $b = 0.2554$, and $c = 1.1228$. (d) Symmetry-resolved operator entanglement $S_{\text{op},S_z}$ in the magnetization sector $S_z = 0$ as a function of time for various $\gamma$ [see (b)]. The black dashed line indicates the logarithmic growth of $S_{\text{op},S_z=0}$ at late times (log-scale time axis). Here the results are converged for time step $\delta tJ = 0.5$ and maximum bond dimension $\chi = 50000$.

Fig. 3(b). Obviously, the classical Shannon entropy of the probabilities in magnetization sectors, $-\sum_{S_z} p_{S_z} \log_2 p_{S_z} \sim \log_2 \delta$, also cannot capture the whole prefactor $\eta$ of the logarithmic growth of operator entanglement at late times; cf. Fig. 1(d).

To identify the behavior of $S_{\text{op},S_z}$, we first consider the difference of symmetry-resolved operator entanglement between the finite magnetization sector and $S_z = 0$ sector by defining $\Delta S_{\text{op},S_z} \equiv S_{\text{op},S_z} - S_{\text{op},S_z=0}$. The results for $\gamma = 0.05J$ are presented in Fig. 3(c), where as the time increases $\Delta S_{\text{op},S_z}$ decays to very small values. This suggests that the symmetry-resolved operator entanglement $S_{\text{op},S_z}$ will take the same value at long times for all of the magnetization sectors. Moreover, we find that $\Delta S_{\text{op},S_z}$ scaled by $1/S_z^2$ collapses almost onto each other at late times for various magnetization sector $S_z$ and can be captured by the function $(a + btJ)^{-c}$ with $c \approx 1$; see the insert in Fig. 3(c). As the contributions of the symmetry-resolved operator entanglement can be rewritten as $\sum_{S_z} p_{S_z} S_{\text{op},S_z} = S_{\text{op},S_z=0} + \sum_{S_z \neq 0} p_{S_z} \Delta S_{\text{op},S_z}$, where the second term $\sim (tJ)^{2\alpha-c}$ vanishes at long times, this suggests that the late-time behavior of the averaged symmetry-resolved operator entanglement can be captured by $S_{\text{op},S_z=0}$, which, just like the total operator entanglement $S_{\text{op}}$, after the initial rise and fall, increases again in a logarithmic manner at late times; see Fig. 3(d). Together with the classical Shannon entropy

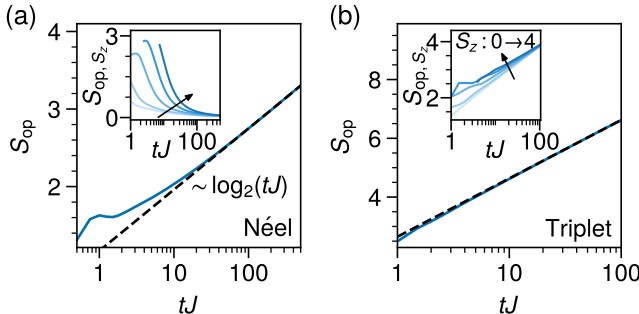

Figure 4: Operator entanglement for quantum dynamics with symmetry being broken to U(1). We consider the Néel initial state in (a) and the product initial state of triplet pairs in (b). The black dashed lines indicate the logarithmic growth of operator entanglement at late times (log-scale time axis). The inserts show the time evolution of symmetry-resolved operator entanglement $S_{\mathrm{op},S_z}$ for $S_z = 0, 1, 2, 3, 4$ (from light to dark). Here we set $\gamma = 0.5J$. The time step $\delta t J$ is 0.25 for (a) and 0.5 for (b). We choose the maximum bond dimension $\chi$ as 4000 for (a) and 12000 for (b).

of the probability distribution $p_{S_z}$, this explains the long-time behavior of the total operator entanglement in our SU(2)-symmetric dissipative quantum dynamics.

Since the operator entanglement dynamics in our SU(2)-symmetric case can be fully understood from the corresponding U(1) subsymmetry, the above results show evidence that the logarithmic growth of operator entanglement at long times is a generic behavior of the dissipative quantum many-body dynamics with U(1) as the symmetry or subsymmetry. This behavior also holds for more broad dissipations beyond dephasing, which is valid even for the open quantum systems with only U(1) symmetry, as we demonstrated below by breaking the symmetry of our quantum dynamics to U(1).

## 5.4  Symmetry breaking from SU(2) to U(1)

We consider the quantum dynamics starting from the Néel state $|\psi_0\rangle = \bigotimes_i |\uparrow\rangle_{2i-1} |\downarrow\rangle_{2i}$ and the product state of triplet pairs $|\psi_0\rangle = \bigotimes_i (|\uparrow\rangle_{2i-1} |\downarrow\rangle_{2i} + |\downarrow\rangle_{2i-1} |\uparrow\rangle_{2i})/\sqrt{2}$, which break the SU(2) symmetry to U(1) at the level of initial states. The results for dissipation strength $\gamma = 0.5J$ are presented in Fig. 4. For both initial states, the operator entanglement $S_{\mathrm{op}}$ shows the logarithmic growth behavior at late times, although the symmetry-resolved operator entanglement $S_{\mathrm{op},S_z}$ decays to small values at late times for the Néel initial state [see the insert of Fig. 4(a)] while it increases logarithmically for the product initial state of triplet pairs as in the SU(2)-symmetric case [see the insert of Fig. 4(b)]. Since the dissipation in our model (1) is proportional to the dipole interaction between neighbor sites, this demonstrates that the logarithmic growth behavior of operator entanglement at late times holds for more broad dissipations beyond dephasing even for the open quantum many-body dynamics with only U(1) symmetry.

## 6  Conclusion

In conclusion, we have studied the far-from-equilibrium dynamics of operator entanglement in a dissipative quantum many-body system with SU(2) symmetry. We find that after the initial rise and fall, the operator entanglement increases again in a logarithmic manner at late times. This behavior can be fully understood from the corresponding U(1) subsymmetry by considering the symmetry-resolved operator entanglement, which, unlike the U(1)-symmetric

case with dephasing, also contributes to the growth of operator entanglement in addition to the classical Shannon entropy associated with the probabilities for the half system being in different symmetry sectors. Our results show evidence that the logarithmic growth of operator entanglement at late times is a generic behavior of dissipative quantum many-body dynamics with U(1) as the symmetry or subsymmetry and for more broad dissipations beyond dephasing. In the future, it would be interesting to further test this conjecture by investigating more symmetries like SU($N > 2$), SO($N$), and SP($2N$). Moreover, in addition to the strong symmetries considered so far, it would be meaningful to study how the presence of weak symmetries impacts the entanglement dynamics of open quantum systems [62, 63].

# Acknowledgements

We implement the SU(2) symmetry of tensors using the package `TensorKit.jl` [64]. We acknowledge support from: ERC AdG NOQIA; MCIN/AEI (PGC2018-0910.13039/501100011033, CEX2019-000910-S/10.13039/501100011033, Plan National FIDEUA PID2019-106901GB-I00, Plan National STAMEENA PID2022-139099NB-I00 project funded by MCIN/AEI/10.13039/501100011033 and by the "European Union NextGenerationEU/PRTR" (PRTR-C17.I1), FPI); QUANTERA MAQS PCI2019-111828-2); QUANTERA DYNAMITE PCI2022-132919 (QuantERA II Programme co-funded by European Union's Horizon 2020 program under Grant Agreement No. 101017733), Ministry of Economic Affairs and Digital Transformation of the Spanish Government through the QUANTUM ENIA project call – Quantum Spain project, and by the European Union through the Recovery, Transformation, and Resilience Plan – NextGenerationEU within the framework of the Digital Spain 2026 Agenda; Fundació Cellex; Fundació Mir-Puig; Generalitat de Catalunya (European Social Fund FEDER and CERCA program, AGAUR Grant No. 2021 SGR 01452, QuantumCAT \U16-011424, co-funded by ERDF Operational Program of Catalonia 2014-2020); Barcelona Supercomputing Center MareNostrum (FI-2023-1-0013); EU Quantum Flagship (PASQuanS2.1, 101113690); EU Horizon 2020 FET-OPEN OPTOlogic (Grant No. 899794); EU Horizon Europe Program (Grant Agreement 101080086 – NeQST), ICFO Internal "QuantumGaudi" project; European Union's Horizon 2020 program under the Marie-Sklodowska-Curie grant agreement No. 847648; "La Caixa" Junior Leaders fellowships, "La Caixa" Foundation (ID 100010434): CF/BQ/PR23/11980043. Views and opinions expressed are, however, those of the author(s) only and do not necessarily reflect those of the European Union, European Commission, European Climate, Infrastructure and Environment Executive Agency (CINEA), or any other granting authority. Neither the European Union nor any granting authority can be held responsible for them.

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
