# Peer review of "Operator entanglement in $\mathrm{SU}(2)$-symmetric dissipative quantum many-body dynamics"

_SciPost Physics_

## Round 1 · Referee Report · Anonymous (Referee 1) · 2024-12-13

Strengths

1- Analyzing operator entanglement growth in this specific dissipative model with SU(2) symmetry is new.

2- The presentation of the results is clear and the manuscript is well written.

Weaknesses

1- The manuscript provides interesting new numerical evidence for a specific model, but no new analytical or universal understanding is achieved.

2- To judge the validity of the results a discussion on details of potential numerical errors for the data is missing. Convergence plots are missing.

Report

In this manuscript the author analyzes the long-time growth of operator entanglement in a dissipative spin-chain with SU(2) symmetry. This follows up earlier works with U(1) symmetric models, where a universal log-growth behavior was attributed to the classical entropy growth stemming from symmetry block diffusion. Here, new numerical evidence suggests that also in this specific SU(2) symmetric setup, the U(1) subsymmetry leads to log-growth behavior.

Almost all general acceptance criteria are fulfilled:

1- The paper is well written and results are clearly presented (only minor suggestions).

2- Details on all physical and numerical parameters are clear.

3- Relevant literature is properly cited.

4- The conclusion is written well and results are objectively summarized

5- The introduction to the problem is well done.

6- Some more details on the numerical convergence behavior are missing and convergence plots should be provided.

While the results are certainly an interesting addition to the existing knowledge, unfortunately I don't agree with the author on the claimed expectations for SciPost Physics being fulfilled:

1- This work is focused on a sub-field of entanglement dynamics in open quantum spin-chains. I fail to see any synergetic link between research areas.

2- As honestly described in the manuscript, the numerical calculations lead to some follow-up insight, amending previous understanding. I don't see it opening up a new research direction. While further investigations, including e.g. SU(N>2) or SO(N) symmetries would indeed be interesting, they would not be multi-pronged, and not be a consequence of this numerical paper.

3- While being interesting, I don't find the findings ground-breaking. As the author honestly writes, this work provides new numerical evidence on operator entanglement growth being a generic feature of U(1). However, this conjecture is not new, and also the paper does not add any new analytical ground-breaking understanding on setups with additional symmetries.

Overall, I therefore think that after some revisions (see below), this paper is publishable, but not in SciPost Physics. All criteria for SciPost Physics Core are fulfilled, so the paper could be published there.

Requested changes

1- Convergence plots need to be shown and a discussion on potential numerical errors/artifacts needs to be added. For example, in Fig. 1(d) results for $\gamma=0.05J$ and $\gamma=0.1J$ exhibit strange wiggles. Is this due to: Fitting problems over finite time? Time-step in the simulations? Finite bond-dimension? The 4-th order Trotter decomposition usually allows to use large time-steps, but the reader should be convinced about the convergence of the results in the time-step and in the bond-dimension by showing some convergence plots in an appendix.

2- (minor) I find some wording in the introduction of the paper a bit "over-the-top". I suggest to remove words like "fantastic", "extremely".

3- Also some statements are a bit imprecise. For example, I would not call operator entanglement a "measure" for classical simulability (strictly speaking, low von Neumann entanglement entropy does e.g. not guarantee an efficient classical state representation [24]). Furthermore, fast operator entanglement growth make simulations with matrix product density operators hard, but not necessarily with another numerical technique. Lastly, I would suggest to use the acronym MPDO (matrix product density operator) instead of MPO, as the latter is more commonly used in a general context and for Hamiltonian representations.

Recommendation

Accept in alternative Journal (see Report)

  • validity: high
  • significance: ok
  • originality: ok
  • clarity: good
  • formatting: good
  • grammar: excellent

Author:  Lin Zhang  on 2025-05-21  [id 5500]

(in reply to Report 1 on 2024-12-13)

Our response to this Report is contained in the attached file.

Attachment:

Response_to_the_Referee_1.pdf

---

## Round 1 · Referee Report · Anonymous (Referee 2) · 2024-12-22

Strengths

-interesting SU(2)-symmetric setup to study operator entanglement
-accurate numerical data

Weaknesses

no physical insight for the prefactors of the operator entanglement growth

Report

The author investigates the operator entanglement entropy in a dissipative system in the presence of SU(2) symmetry, considering dissipation beyond dephasing. This work makes a valuable contribution to the study of open quantum systems by extending previous findings on U(1)-symmetric systems to more complex symmetry structures and dissipative mechanisms. The results provide valuable insights into the interplay between symmetry, dissipation, and entanglement dynamics, which are of substantial interest in the field of quantum many-body physics and classical simulability of quantum systems. The analysis is rigorously performed and the results are clearly presented. However, the paper is lacking an interpretation of the logarithmic growth of the operator entanglement, as also stressed in my comments below. For this reason I believe that the paper is more suitable for Scipost Physics Core.

Suggestions:

a) The claim of non universality of the prefactor \eta of the logarithmic growth in Fig. 1 appears not completely justified. In fact it could be that the putative non universality can be attributed to the finite times that can be reached with the numerics. I would suggest to rephrase the discussion leaving the scenario of universality open.

b) In Fig. 2 it is not clear what is the prefactor of the log growth. Is it the same for both the number entropy and the entanglement entropy?

c) From fig 3 it is not clear what would be the growth of the number entropy? Also, what is the prefactor of the log growth?
Similarly, in Fig.4 there is no discussion of the prefactor of the log growth.

Recommendation

Accept in alternative Journal (see Report)

  • validity: high
  • significance: high
  • originality: good
  • clarity: high
  • formatting: perfect
  • grammar: excellent

Author:  Lin Zhang  on 2025-05-21  [id 5501]

(in reply to Report 2 on 2024-12-22)

Our response to this Report is contained in the attached file.

Attachment:

Response_to_the_Referee_2.pdf

---

## Round 1 · Referee Report · Anonymous (Referee 3) · 2025-1-2

Strengths

  1. Interesting numerical observation about operator entanglement in open quantum systems with strong symmetry

  2. Clear and well presented

Weaknesses

  1. Lack of analytical insights

Report

Inspired by the results of Ref. [38] for an open quantum system with U(1) 'strong symmetry', the author studies the evolution of operator entanglement in a spin chain with SU(2) 'strong symmetry' under Lindblad dynamics. The model is a Heisenberg spin chain, with incoherent nearest-neighbor projections on singlets, which preserve the SU(2) symmetry. The initial state is chosen such that it is SU(2) symmetric (except in Section 5.4, where the initial state breaks the SU(2) symmetry, and is only U(1)-symmetric). The model is thoroughly studied numerically.

The main observation is that the operator entanglement undergoes an initial linear growth, followed by a decrease caused by decoherence, and a slow logarithmic growth at late times. This is the same phenomenology as in the U(1) case of Ref. [38], however, as emphasized by the author, an important difference with respect to the U(1) case is that the operator entanglement at long times does not come exclusively from the Shannon entropy of the number fluctuations between the to subsystems. This is reflected in the 'symmetry-resolved' operator entanglement which no longer vanishes, contrary to what happened in the U(1) case of Ref. [38].

The numerical results are presented in a clear way and are well discussed. Unfortunately, not much analytical insights are given, and the main observation of the paper remains unexplained.

Yet, I think the results are interesting enough, and could motivate further work in that direction. Therefore, in my opinion, it meets the Scipost Physics acceptance criterion 'Detail a groundbreaking theoretical/experimental/computational discovery'. I am not sure it meets any of the other criteria though.

Requested changes

  1. The 'symmetry-resolved' operator entanglement should be defined carefully. The author gives a very brief discussion of that definition, and cites a number of references (many of which are about state entanglement, not operator entanglement, which might cause confusion), as if the author believed that there is a unique way of defining 'symmetry-resolved operator entanglement'. Let me stress that this is not the case. Here the author uses a definition similar to the one of Ref. [38], but this choice should be discussed and justified. Other definitions of 'symmetry-resolved operator entanglement' are being used in the literature, for instance in Ref. [59], that do not match the one of Ref. [38].

  2. When the author introduces the model, they refer to Refs. [47,48] as papers that considered a similar model. These works were about superdiffusion and possible appearance of KPZ scaling in spin dynamics. Are there any connections between this work and superdiffusion/KPZ scaling? If so, this would be extremely interesting. But this is not discussed at all. Would the author be able to comment on this?

  3. In Ref. [38], an analytical undersanding for the logarithmic growth is reached by looking at the strong-dephasing limit, where the model maps to a classical stochastic model. Could something like this be done here in the SU(2) case? If so, could this provide insights into the observed phenomeon?

  4. The author cites Refs. [41,42,43] and mentions the fact that symmetries can enhance entanglement of stationary states of systems with strong symmetries. But couldn't such a mechanism be invoqued here to explain the non-vanishing symmetry-resolved operator entanglement at long times? If not, why? It would be interesting if the author could elaborate on the connections or differences with these works.

Some weird sentences (mostly in the introduction) could be rephrased: - 'three-rank tensor' -> 'rank three tensor' - 'increasement' -> 'increase' - 'Not only being useful in studying the closed systems' - 'which is dubbed as the operator entanglement' - 'it was recently studied that' - 'we use MPO to represent the density matrix'

Recommendation

Ask for minor revision

  • validity: high
  • significance: good
  • originality: good
  • clarity: high
  • formatting: excellent
  • grammar: good

Author:  Lin Zhang  on 2025-05-21  [id 5502]

(in reply to Report 3 on 2025-01-02)

Our response to this Report is contained in the attached file.

Attachment:

Response_to_the_Referee_3.pdf

---

## Editorial Decision

resubmitted